# MiRNA-146a—A Key Player in Immunity and Diseases

**DOI:** 10.3390/ijms241612767

**Published:** 2023-08-14

**Authors:** Irina Gilyazova, Dilara Asadullina, Evelina Kagirova, Ruhi Sikka, Artur Mustafin, Elizaveta Ivanova, Ksenia Bakhtiyarova, Gulshat Gilyazova, Saurabh Gupta, Elza Khusnutdinova, Himanshu Gupta, Valentin Pavlov

**Affiliations:** 1Subdivision of the Ufa Federal Research Centre of the Russian Academy of Sciences, Institute of Biochemistry and Genetics, 450054 Ufa, Russiaelzakh@mail.ru (E.K.); 2Institute of Urology and Clinical Oncology, Department of Medical Genetics and Fundamental Medicine, Bashkir State Medical University, 450008 Ufa, Russiasqwer1@yandex.ru (A.M.); gulshatik2001@mail.ru (G.G.);; 3Department of Biotechnology, Institute of Applied Sciences and Humanities, GLA University, Mathura 281406, India

**Keywords:** miRNA, immunity, cancer, miRNA-146a

## Abstract

miRNA-146a, a single-stranded, non-coding RNA molecule, has emerged as a valuable diagnostic and prognostic biomarker for numerous pathological conditions. Its primary function lies in regulating inflammatory processes, haemopoiesis, allergic responses, and other key aspects of the innate immune system. Several studies have indicated that polymorphisms in miRNA-146a can influence the pathogenesis of various human diseases, including autoimmune disorders and cancer. One of the key mechanisms by which miRNA-146a exerts its effects is by controlling the expression of certain proteins involved in critical pathways. It can modulate the activity of interleukin-1 receptor-associated kinase, IRAK1, IRAK2 adaptor proteins, and tumour necrosis factor (TNF) targeting protein receptor 6, which is a regulator of the TNF signalling pathway. In addition, miRNA-146a affects gene expression through multiple signalling pathways, such as TNF, NF-κB and MEK-1/2, and JNK-1/2. Studies have been carried out to determine the effect of miRNA-146a on cancer pathogenesis, revealing its involvement in the synthesis of stem cells, which contributes to tumourigenesis. In this review, we focus on recent discoveries that highlight the significant role played by miRNA-146a in regulating various defence mechanisms and oncogenesis. The aim of this review article is to systematically examine miRNA-146a’s impact on the control of signalling pathways involved in oncopathology, immune system development, and the corresponding response to therapy.

## 1. Introduction

miRNAs are a class of small non-coding RNA molecules that play an essential role in regulating numerous pathways and important biological processes including innate immunity, inflammatory responses, haematopoiesis, development of malignancies and metastases. These short, single-stranded RNA molecules measure between 18–24 nucleotides in length. Their physiological role is to monitor gene expression at the post-transcriptional level by binding to the target mRNA’s 3’-untranslated region. This binding can lead to the degradation of the mRNA or block its translation [1].

miRNAs play a critical role in gene regulation, affecting approximately 33% of human genes [2]. miRNAs exhibit unique patterns of modulation in inflammatory cells, cancer cells, and other cells and tissues associated with specific pathological disorders. Dysregulation of miRNAs, either up or downregulation, often leads to functional abnormalities in cellular activity. Therefore, these nucleic acid components have great potential as diagnostic biomarkers and clinical predictors, in particular for inflammatory diseases. To gain insights into the molecular mechanisms underlying the pathogenesis of inflammatory diseases, extensive studies have been conducted. These investigations aim to unravel the intricate involvement of miRNAs in various pathological conditions. The comprehensive understanding of these mechanisms will not only facilitate the development of new biomarkers but also uncover novel therapeutic targets.

## 2. MiRNA-146a—Function, Role, Involvement in Innate Immunity, Development of Inflammatory Responses, and Haematopoiesis

miRNAs play crucial roles in immune responses, supported by evidence of selective miRNA expression in immune cells [3]. miRNA-146a is located on chromosome 5 and has been extensively studied. Knocking out miRNA-146a in mice resulted in autoimmunity, increased sensitivity to lipopolysaccharides (LPS), and heightened pro-inflammatory activity upon endotoxin exposure. Additionally, miRNA-146a deficiency in mice led to age-related tumour formation and myeloproliferation, suggesting its role in regulating immune cell proliferation [4]. This implied that miRNA-146a is essential for transmitting nuclear factor kappa B (NF-κB) signals, and its absence contributed to myeloid malignancies [5]. Similarly, miRNA-146a-deficient mice mimic myelodysplastic syndrome (MDS) patients with fifth chromosome deletion.

Toll-like receptors (TLRs) are crucial for innate immunity as they recognize microbial structures and initiate immune responses. Humans have 11 TLRs (TLR-1 to TLR-11), expressed in different cell types, capable of binding various ligands [6]. TLRs activate the innate immune response through the MyD88 signalling pathway or TRIF signalling pathway. miRNA-146a targets key signalling proteins in the MyD88-dependent pathway, demonstrating its significance in innate immunity (Figure 1). 

Adapter molecules activate downstream transcription factors, including NF-κB, mitogen-activated protein kinase (MAPK), interferon regulatory factor family members (such as IRF3 and IRF7), and activator protein-1 (AP-1), which induce proinflammatory cytokines, type 1 interferons (IFNs), and antiviral proteins [7]. Most TLRs use the MyD88 pathway, which activates NF-κB and MAPK, except for TLR3, which can activate the TRIF signalling pathway [8]. Sufficient TLR signalling is necessary to effectively kill the pathogen, but an uncontrolled TLR reaction can harm the host. Thus, the TLR-mediated inflammatory response should be supervised, and the complex regulatory potential provided by miRNAs, especially miRNA-146a, is crucial for the physiological functioning of this type of innate immune response.

miRNA-146a targets adapter proteins interleukin-1 receptor-associated kinase 1 (*IRAK1*; kinase associated with the IL-1 receptor 1) and *TRAF6* (factor associated with the TNF receptor 6) involved in TLR and IL-1 receptors signalling, inhibiting proinflammatory mediator secretion, and blocking TLR signals [9]. This suggests that targeting adapter molecules with miRNA-146a might regulate TLR-mediated innate immunity [9].

According to the literature, miRNA-146a plays an important role in endotoxin-induced tolerance [10]. This mechanism decreases monocyte responsiveness to LPS after prolonged or repeated exposure, preventing inflammation from continuous contact with bacterial components. LPS exposure increases miRNA-146a levels in THP-1 cells (a human leukaemia monocytic cell line), which negatively correlates with TNF levels, indicating tolerance to LPS [10]. Positive miRNA-146a regulation is necessary for tolerance activation, and exogenous miRNA-146a transfection induces tolerance even without LPS priming, while miRNA-146a knockdown reduces tolerance to LPS [11]. miRNA-146a also contributes to innate immune tolerance in new-borns’ intestines, preventing intestinal epithelial cell apoptosis upon bacteria exposure [12]. These studies indicate that although miRNA-146a is induced by TLR signalling, its primary role may be to serve as a dominant regulator of negative feedback to prevent uncontrolled inflammation resulting from prolonged contact with bacterial agents.

miRNA-146a is extensively studied in inflammation and immunity, acting as a dominant negative feedback regulator in vertebrate innate immune responses. Mechanistically, miRNA-146a targets key components of the MyD88-dependent signalling pathway, such as IRAK1 and TRAF6, leading to a coordinated decrease in the synthesis of various inflammatory mediators. Although most of the research has focused on the TLR pathway and inflammatory cells, miRNA-146a is expressed in various other cell types and may participate in other functions depending on the context (Figure 2).

In haematopoiesis, miRNA-146a plays a vital role, regulating target gene expression and contributing to certain hematopoietic diseases. It is predominantly expressed in multipotent stem cells, granulocytic/monocytic progenitors, megakaryocytic/erythroid progenitors, and myeloid progenitors. Overexpressing miRNA-146a in hematopoietic bone marrow stem cells increases granulocytes and red blood cells but worsens lymphopoiesis. Forced miRNA-146a expression impairs bone marrow repair [13]. Healthy individuals show high miRNA-146a expression in bone marrow CD34 precursors, but patients with acute myeloid leukaemia have lower levels, particularly in monocytes, granulocytes, erythrocytes, and megakaryocytes from both peripheral blood and bone marrow of healthy donors [14]. Differentiation of the acute promyelocytic leukaemia cell line NB4 leads to decreased miRNA-146a expression upon trans-retinoic acid induction [15]. However, miRNA-146a is actively expressed in regulatory T cells and type 1 T helper cells compared to mature T cells and type 2 T helper cells. miRNA-146a expression in megakaryocytes varies across different experiments [16,17,18]. The role of miRNA-146a in megakaryocyte development warrants further investigation, considering its diverse effects in different experimental contexts.

miRNA-146a targets *TRAF6* and *IRAK1* genes, leading to increased protein expression in knockout cell lines. *TRAF6* is involved in myeloid cells development, and overexpression associates with acute myeloid leukaemia growth, suggesting that miRNA-146a acts as a tumour suppressor gene. Mice without miRNA-146a expression develop tumours in lymphoid organs, indicating its inhibition promote tumour growth and metastasis. However, miRNA-146a deficiency alone is not sufficient for oncogenesis, indicating that other factors interact with malignant tumour development [19]. Chromosomal translocations and gene mutations in acute myeloid leukaemia (AML) associate with miRNA-146a expression [20]. Recently, an increasing number of studies have proven that microRNA expression can influence the clinical outcome of AML. In particular, a study by Zhang et al. [21] indicates that miRNA-146a is highly expressed in both acute lymphoblastic leukaemias (ALL) and AML in children. Given its ability to regulate gene expression, miRNA-146a may serve as a diagnostic and prognostic marker for hematopoietic diseases. In addition, targeting miRNA-146a expression may be a potential therapeutic strategy for controlling leukemogenesis. Overall, the study of miRNA-146a expression advances our understanding of the molecular mechanisms underlying normal and abnormal haematopoiesis.

## 3. Role of miRNA-146a in the Development of Autoimmune Diseases and Allergic Reactions

The discovery of miRNA-146a as a crucial inhibitor of the innate immune response has greatly aided in understanding the possible involvement of miRNAs in the development of various autoimmune diseases and disorders. It is now assumed that the initial trigger for the development of many autoimmune diseases is chronic inflammation. This chronic inflammation, caused by the constant production of mediators by autoimmune cells, increases the negative reaction of the immune system against its own body and, by the principle of feedback, prevents the completion of the immune response. The literature suggests a significant role for of miRNA-146a in the development of autoimmune diseases such as rheumatoid arthritis (RA), systemic lupus erythematosus (SLE), and Sjogren’s syndrome [22]. 

RA is a prototypical autoimmune disease with multifactorial origins that primarily affects small joints, causing erosive and destructive polyarthritis. This condition is frequently accompanied by the presence of autoantibodies in affected individuals [23]. The existing body of evidence on miRNAs indicates that miRNA-146a is notably upregulated in patients with RA [24]. The hypothesis of increased expression of miRNA-146a, which may have a significant impact on the development of RA, is supported by studies showing that miRNA targets important molecules such as NF-κB and IRAK1, which are involved in osteoclastogenesis by activating osteoclast-derived progenitor cells [25]. Therefore, enhanced expression of miRNA-146a and its subsequent suppression of IRAK1 and NF-κB may potentially mitigate osteoclastogenesis and limit joint damage. Furthermore, elevated expression of miRNA-146a may serve as a potential biomarker for RA. Additionally, TNF can stimulate osteoclastogenesis and induce the expression of miRNA-146a. Thus, overexpression of miRNA-146a could be harnessed to suppress TNF-mediated osteoclastogenesis.

SLE is a complex autoimmune disease characterized by the production of autoantibodies, activation of the complement system, and the deposition of immune complexes, all of which contribute to tissue and organ damage [26]. It is caused by genetic factors that impair immune tolerance and promote the production of autoantibodies. These genes can interact with other genetic factors that signal innate immunity and the production of IFN-I, trigger the release of leukocytes, inflammatory mediators, and autoantibodies into targeted organs.

The presence of elevated serum IF”α le’els and altered expression of IFN-inducible genes are well-established in patients with SLE [27]. IL-12 and IL-23 can activate the IFN receptor α/β (IFNAR), resulting in increased IFN-I secretion. This activation leads to the upregulation of a large number of IFN-I-regulated genes, forming the SLE RNAi-associated signature. Among these genes, *IRF5*, *IRF7*, *ILT3*, and *IFIH1* are SLE -associated genes, and some of them are responsible for regulating the expression of IFN-I and other disease-related cytokines. Together, this “IFN signalling” has become a major contributing factor to the development of SLE pathology.

Genome-wide association studies (GWAS) have identified polymorphic regions in genomic segments for miRNA-146a associated with a predisposition to SLE. In the Asian population, the rs57095329 variant in the promoter region of the miRNA-146a gene, which presumably reduces the efficiency of binding to transcription factors, contributes to decreased miRNA expression and increases the risk of developing SLE [28]. Similarly, an increased risk of SLE has been reported in Europeans with the rs2431697 variant in the intergenic region of the miRNA-146a gene, which contributes to reduced miRNA expression [29]. These studies collectively suggest that insufficient expression of miRNA-146a in SLE may contribute to aberrant IFN signalling during the course of the disease.

Sjogren’s syndrome is a chronic autoimmune disease characterized by a chronic autoimmune and lymphoproliferative process in secreted epithelial glands [30]. In theory, miRNA-146a activation suggests a concomitant reduction in the production of pro-inflammatory cytokines, but the opposite has been reported in patients with this syndrome. This result is possibly due to an unknown mechanism that blocks the recognition ability of miRNA-146a to detect its own set of target genes, causing inadequate clearance of the antigen and subsequent inflammation.

One possible explanation for this mechanism could be the difference in the expression of the miRNA-146a target genes *IRAK1* and *TRAF6* found in patients with Sjogren’s syndrome [30]. In these patients, increased miRNA-146a expression leads to an upregulation of TRAF6 expression but a simultaneous downregulation of *IRAK1* expression. The potential lack of complete concerted repression of upstream NF-κB molecules, in the presence of enhanced expression of miRNA-146a, could potentially explain this novel inflammatory response observed in patients with Sjogren’s syndrome.

Altogether, the reported data provide a valid argument for the role of miRNA-146a in the pathobiology of autoimmune diseases. However, more research is necessary to establish the specific role of miRNA in the actual mechanism of pathogenesis. The potential investigation of circulating miRNA-146a as a diagnostic and/or prognostic biomarker of inflammation remains a therapeutic focus for patients with these pathologies.

In addition to autoimmune diseases, miRNAs are of particular clinical importance in allergic diseases. These conditions are characterized by tissue inflammation and can be particularly difficult to diagnose and characterize, given that the measurement of release mediators, such as cytokines, in the blood is unreliable. Hence, profiling of miRNA expression in blood and other body fluids is becoming essential for identifying and developing new non-invasive disease biomarkers. Allergen-specific immunotherapy is the only known effective treatment for IgE-mediated allergies, and sublingual immunotherapy is a common and safe strategy for addressing inadequate immune response in patients with allergies. 

miRNA-146a may play a role in allergen-specific immunotherapy of allergic diseases by modulating Treg cells, which are central regulators of the immune response. The role of miRNA-146a is well studied in allergic asthma, where it can modulate the expression of inflammation mediators in airway smooth muscle, thus contributing to the pathogenesis of this disease [31]. miRNA-146a serves as a negative regulator of inflammatory gene expression in many cell types, including monocytes, fibroblasts, endothelial cells, epithelial cells [32], and smooth muscle cells of the airways. Numerous studies have described miRNA-146a targets, including cyclooxygenase-2 (COX-2) [33], human antigen R (HuR), IRAK1, TRAF6 [34], IL-1β, IL-6, and IL-8. Among these, IRAK1, TRAF6, IL-6, and IL-8 are the most common targets in human airway smooth muscle cells.

The role of miRNA-146a in rhinovirus-induced airway inflammation and allergic asthma exacerbation is well described by Anet Laanesoo et al., where miRNA-146a suppresses signalling through the NF-κB pathway and inhibits proinflammatory chemokine production in primary human bronchial epithelial cells. These works showed that multiple components, including IRAK1 and the caspase 10 recruitment domain (CARD10), from the NF-κB pathway directly target miRNA-146a [35]. In this regard, miRNA-146a overexpression has been shown to suppress endogenous levels of IRAK1 and CARD10 as well as the NF-κB-inducible chemokine IL-8 and chemokine ligand 1 (CXCL1) in many cell types, including human bronchial epithelial cells. Moreover, while studying monolayer cultures of primary human bronchial epithelial cells, Anet Laanesoo et al. concluded that miRNA-146a suppresses the expression of proinflammatory chemokines and induces interferon response genes during rhinovirus infection in bronchial epithelial cells. Another finding of the authors was that miRNA-146a reduces the number of infected cells and suppresses neutrophil migration during rhinovirus infection by transfection with miRNA-146a mimetics, infection with the green fluorescent protein RV-A16, and subsequent evaluation by fluorescence microscopy or flow cytometry [36].

An interesting study was conducted by Brian S Comer et al., in which they compared miRNA-146a expression in human airway smooth muscle cells from patients without asthma and asthmatics treated with cytomix (IL-1β, TNF-α and IFNγ) and analysed the effect of miRNA-146a on COX-2 and IL-1β expression. As a result, cytomix treatment was found to increase miRNA-146a content. Induction by cytomix was higher than induction by individual cytokines, and asthmatic cells showed higher levels of miRNA-146a expression after cytomix treatment, confirming that miRNA-146a can modulate the expression of inflammatory mediators in airway smooth muscle, thereby contributing to asthma pathogenesis [37].

Thus, it is recommended that miRNA-146a research be expanded to examine its efficacy in screening many autoimmune and allergic diseases to confirm its role as a diagnostic biomarker. This would complement its role in assessing disease severity, monitoring treatment response, and personalized therapy [35].

## 4. miRNA-146a in Infection Diseases

miRNA-146a has been recognized as a miRNA with anti-inflammatory properties. It functions in the regulation of pro-inflammatory immune reactions by limiting excessive stimulation, thereby preserving the immune system’s homeostatic status [8,10,32,38,39,40]. It negatively regulates the inflammatory response by affecting the TRAF6, thereby decreasing the expression of the central immune transcription factor NF-κB [39,40,41,42,43]. miRNA-146a also suppresses the NF-κB-inducible proinflammatory chemokines IL-8, CXCL1, and CCL5 and increases IFN-λ production [44,45]. In addition, miRNA-146a is engaged in the regulation of TLR receptors recognizing pathogen-associated molecular patterns (PAMPs) [44,45,46,47,48,49,50,51,52,53].

miRNA-146a also plays a role in T cell subpopulations. It is differentially expressed in Th1 and Th2 cells. An increased percentage of INF-γ-producing T cells is observed in mice with reduced expression of miRNA-146, although the miRNA-146 expression level in CD4+ T cells is usually relatively low. However, it is abundantly expressed in human memory T cells. Moreover, miRNA-146a is stimulated upon T-cell receptor (TCR) induction, and its expression depends on NF-κB induction [54]. The promoter of the gene encoding miRNA-146a has several binding sites for the nuclear factor alpha B transcription factor (NF-κB), which plays a critical role in the regulation of immune response [8,55]. miRNA-146a transcription is managed by NF-κB, even though these two molecules—IL-6 and miRNA-146a—perform opposite functions in the regulation of the inflammatory process [56]. Recent research has indicated the critical role of miRNA-146a in macrophage polarization [45,46]. For example, Hung et al. indicated that miRNA-146a can reduce the proinflammatory response of M1 macrophages by promoting M2 polarization [57].

The role of miRNA-146a has been widely known in bacterial infections and viral infections, particularly mycobacteria [8,58,59,60,61,62,63]. Recent studies have found that miRNA-146a can block the mRNA sequence of chemokine ligand 5 (CCL5) and reduce its release by HIV-infected macrophages, thereby affecting the migration of monocytes. The involvement of monocytes is necessary for effective control and clearance of the virus or other viral infections. This indicates that miRNA-146a is a negative post-transcriptional regulator of CCL5 production [64]. CCL5, also known as regulated upon activation, expressed and secreted by normal T cells (RANTES), recruits monocytes from the bloodstream into tissue. HIV-1 infection has also been demonstrated to induce CCL5 production and sustained accumulation of miRNA-146a in macrophages. Increased levels of miRNA-146a reduce CCL5 secretion, resulting in decreased monocyte migration to sites of infection [65].

An association between aberrant expression of miRNA146a and various pathological conditions has been consistently reported. For example, there is evidence that miRNA-146a is induced by at least 18 single-stranded RNA (ssRNA) or double-stranded DNA (dsDNA) neurotropic viruses, including SARS-CoV-2, the causative agent of COVID-19 [66].

Interestingly, Notch-1 has been determined as a potential miRNA-146a target gene [67]. miRNA-146a overexpression significantly reduced Notch-1 levels and subsequently reduced IL-6 production in the activated LPS macrophage cell line RAW264.7 [68]. miRNA-146a levels were significantly elevated in cells infected with coxsackievirus B type 3 [69]. miRNA-146a overexpression is also observed in Crohn’s disease and ulcerative colitis [70]. miRNA-146a expression is induced and TRAF6 and STAT-1 expression, as well as ROS (reactive oxygen species) activity, are suppressed in human microglial cells upon infection with Zika virus NS1. miRNA-146a overexpression led to suppression of proinflammatory and cellular antiviral responses in microglial cells [71]. Patients with haemorrhagic fever with renal syndrome (HFRS) also had elevated levels of lysosomal miRNA-146a [72]. 

Other studies have demonstrated that hepatitis A virus (HAV) infection raises miRNA-146a expression, which in turn partially inhibits the induction of IFN-β synthesis, thereby promoting viral replication. Mechanistically, miRNA-146a targets and degrades TRAF6, a key adaptor protein in the RIG-I/MDA5-mediated IFN-I signalling pathway. Because TRAF6 is required for induction of IFN-β, inhibition of this protein attenuates IFN-β signalling. HAV disrupts RIG-I/MDA5-mediated IFN-I signalling in part through cleavage of the important adaptor molecule TRAF6 by miRNA-146a [45]. In another study, miRNA-146a deficiency contributed to the differentiation of Kupffer cells towards a pro-inflammatory phenotype. The persistence of the hepatitis B virus or the treatment of surface antigen virus significantly increased miRNA-146a expression and thereby disrupted the transition of Kupffer cells into a pro-inflammatory phenotype [73]. It has also been reported that miRNA-146a deletion can increase the antiviral response in cells infected with HIV-1 [74].

Patients with COVID-19 showed elevated levels of IL-6 and downregulation of miRNA-146a compared to healthy individuals of comparable age. This indicates an imbalance in the physiological axis of IL-6/miRNA-146a in the pathogenesis of SARS-CoV-2 infection [75,76,77]. The imbalance in the IL-6/miRNA-146a axis may depend, at least in part, on the fact that IL-6 stimulates nuclear factors other than NF-κB or acts synergistically with NF-κB, thereby dramatically enhancing IL-6 synthesis without concomitant induction of miRNA-146a transcription. A similar pattern has been observed in sepsis [78].

miRNA-146a plays a key role in maintaining homeostasis during *A. fumigatus* infection and in repressing the translational of pro-inflammatory genes in macrophage-like THP-1 cells. This is evident as miRNA-146a downregulates the production of IL-6 and TNF in response to A. fumigatus [79], serving as a negative feedback regulator to limit excessive activation of innate immune cells and minimize tissue damage by preventing excessive inflammation [80].

It has been observed that miRNA-146a is involved in bacterial infections, where it induces cell wall endotoxin through TLR4 stimulation in various cell types [81,82]. Moreover, there is evidence that extracellular miRNA-146a can induce cross-tolerance to LPS at the posttranslational level through degradation of proteosomal IRAK-1. This extracellular miRNA-146a is released from activated macrophages and transported to endothelial cells, resulting in a decrease in endothelial permeability and stimulation of acute inflammatory response [83]. 

Notably, the level of circulating miRNA-146a decreases significantly with age and in the presence of chronic age-related diseases and conditions [84,85]. Depletion of miRNA-146a leads to overproduction of IL-1, IL-6, and TNF and it is also associated with the signalling pathway of toll-like receptors and act as a controller of transcription factors TGFB1 and IL-1B [86,87].

miRNA-146a expression is also connected with biochemical parameters, including NT-proBNP and the absolute number of neutrophils. Additionally, miRNA-146a directly controls thrombo-inflammatory processes by inhibiting the NF-κB pathway and pro-inflammatory components [8,10,38,39,40,41,42,43]. This miRNA is predominantly expressed in cells contributing to thrombosis, such macrophages, neutrophils, endothelial cells, and platelets [88]. Among other functions, it has been demonstrated that miRNA-146a downregulation in pneumonia patients correlates with a higher risk of adverse cardiovascular events due to the aggravation of inflammatory and prothrombotic reactions related to severe COVID-19 [89]. Dysregulation of miRNA-146a expression and its targets is also associated with functional genetics polymorphisms in the miRNA-146a sequence [90]. 

Single nucleotide polymorphisms (SNPs) rs2910164 and rs57095329 in the miRNA-146a gene sequence are often found to be associated with different diseases. They regulate mature miRNA-146a expression profiling and are associated with various cancers, inflammatory, and infectious diseases [62,91,92,93,94,95,96]. The rs57095329AG and rs2910164GC in the gene promoter reduce mature miRNA-146a expression and silence its target genes by interfering with pre-miRNA processing [30,90]. Various studies have reported that SNP rs2910164 of miRNA-146a is related to many pathological conditions, including autoimmune diseases, severe sepsis, tuberculosis, cancer, leprosy, hepatitis B, intestinal metaplasia, and dysplasia in individuals infected with *Helicobacter pylori* [97,98,99,100,101,102,103,104]. SNP rs2910164GC is noted to be concerned with a hyperactive response of the immune system and an increased contamination risk [101].

The association between miRNA-146a and malaria has also been established [105,106]. Moreover, in primigravid Ghanaian women, the occurrence of pregnancy-associated malaria was found to be more likely due to carrying of miRNA-146a rs2910164GC and rs2910164CC, indicating the potential involvement of miRNA-146a in this complication [105]. The researchers of the study suggested that miRNA-146a may play a role in the innate component of protective malarial immunity.

## 5. Participation of miRNA-146a in the Pathogenesis of Malignant Tumours of Various Localizations

The association between miRNAs and malignant tumours was first noted by Calin et al. in patients with chronic lymphocytic leukaemia, where abnormal expression of miRNA-15 and miRNA-16 was observed [107]. Subsequently, numerous studies have been conducted to identify biologically and clinically relevant miRNA expression patterns. Several studies carried out in papillary thyroid carcinoma were among the first to show miRNA-146a abnormalities in some types of onco-pathology. For example, He et al., in 2005, showed that miRNA-146a is activated with the highest frequency compared to normal thyroid tissue [108]. The increased expression of miRNA-146a in anaplastic thyroid cancer may be related to the high and spontaneous NF-κB activity observed in these tumour cells. In fact, NF-κB was shown to stimulate miRNA-146a transcription [109]. Studies have also been carried out on abnormalities of miRNA-146a expression in pancreatic carcinoma. It was found that miRNA-146a expression was reduced in pancreatic cells compared to normal tissue [95]. Re-expression of miRNA-146a in pancreatic cancer cells led to inhibition of both *EGFR* and NF-κB signalling. This effect is accompanied by a decrease in the invasiveness of these cancer cells [110]. miRNA-146a expression was also studied on breast pathologies. One study showed how the breast cancer metastasis suppressor 1 (BRMS1) can act by increasing miRNA-146a levels [111]. BRMS1 is a metastasis suppressor that affects several stages of the breast carcinoma metastasis cascade, acting through different mechanisms involving different genes whose function is related to the control of metastatic potential. Thus, BSMS1 has been shown to markedly activate the expression of miRNA-146a and miRNA-146b in breast carcinoma cells and, through this mechanism, inhibit the expression of *EGFR*, leading to a reduction in the metastatic potential of these cells.

After conducting similar studies, scientists became interested in the molecular mechanisms responsible for the downregulation of miRNA-146a. They revealed that inactivated tumour suppressor genes in cancer are often associated with the mechanism of RNA methylation at CpG sites in promoter regions [112]. For example, this phenomenon contributes to the reduction of miRNA-146a expression in several non-small cell lung cancer cell lines [113]. Similarly, a hypermethylated miRNA-146a promoter has been detected in hepatocellular carcinoma [114] and prostate cancer cells [115]. Interestingly, Liu et al. observed that in prostate cancer cells, lncRNA PVT1 can induce CpG methylation in the miRNA-146a promoter [115]. Besides methylation, other epigenetic mechanisms can also affect miRNA-146a transcription.

Thus, miRNA-146a was found to play a complex role in oncogenesis, which implicates cancer development and progression. Its role in cancer development can vary depending on the specific tumour type, conditions, and tumour microenvironment. The same miRNA can have different effects depending on specific cellular and molecular environment conditions. In some cases, miRNA-146a functions as a tumour suppressor, while in other cases it may act as an oncogene. Direct miRNA-146a target genes become overexpressed, which contributes to increased cell proliferation, invasion, metastasis, and cell survival. Dysregulation of miRNA-146a expression has been observed in various types of oncopathology, thus further studies are needed to better understand the exact mechanisms by which miRNA-146a promotes oncogenesis and to explore its potential as a therapeutic target in oncology therapy.

### 5.1. miRNA-146a as a Tumour Suppressor

To date, numerous studies have explored the role of miRNA as a tumour suppressor, focusing on key aspects of tumour cell transformation prophylaxis. One such aspect is inflammatory signalling, where miRNA-146a acts as a negative regulator. It targets and inhibits key molecules involved in inflammatory pathways, such as kinases associated with interleukin-1 receptor and TRAF6. By inhibiting these pro-inflammatory factors, miRNA-146a helps prevent chronic inflammation, which is associated with an increased risk of developing cancer [116].

Another essential role of miRNA-146a as a tumour suppressor is in cell proliferation and survival. It has been found that miRNA-146a inhibits cell proliferation and promotes apoptosis in malignant cells. It influences genes involved in cell cycle regulation, such as cyclin-dependent kinase 2 (CDK2) and cyclin D1, thereby inhibiting tumour cell growth [117]. Evidence suggests that miRNA-146a may affect apoptosis in human glioblastoma. Treatment of U-87 MG cells with curcumin, a natural compound, induced the expression of miRNA-146a. Combined treatment of cells with curcumin and an anticancer drug suppressed proliferation and induced apoptosis to a greater extent than therapy with temozolomide alone. However, repeating the experiment in the presence of the miRNA-146a inhibitor prevented this increase. Suppression of NF-κB signalling is central to the role of miRNA-146a in glioblastoma [118]. Recent work also indicates the involvement of transforming growth factor-beta (TGF-β) in this mechanism. Overexpression of miRNA-146a in microglial cells suppressed maternal factor levels against decapentaplegic homolog 4 (SMAD4) and consequently matrix metalloproteinase 9 (MMP9) [119].

In addition to this, invasion and metastasis are also a major aspect. Studies have shown that miRNA-146a can inhibit the invasive and metastatic potential of cancer cells by affecting genes related to epithelial–mesenchymal transition (EMT), a process that promotes invasion and metastasis of cancer cells. For instance, a preliminary experiment compared the expression of miRNA-146a in 62 samples of oesophageal squamous cell carcinoma and corresponding non-tumoural tissues. In malignant tissues, miRNA-146a levels were significantly reduced, and this reduction correlated with poor overall survival and progression-free survival in patients [120]. Less than a year later, the same group of researchers published the results of further experiments, concluding that miRNA-146a can inhibit the progression of epithelial–mesenchymal transition in oesophageal cancer cell lines. Treatment of cells with the miRNA-146a inhibitor resulted in decreased levels of E-cadherin protein as well as increased levels of Snail and Vimentin proteins. Notch2 was also found to be a direct inhibitor of miRNA-146a in these cells [120]. Another role of miRNA-146a as an oncosuppressor lies in immune evasion, where it modulates the immune response in the tumour microenvironment. miRNA-146a can target immune checkpoint molecules, such as programmed cell death ligand 1 (PD-L1), and inhibit their expression, thereby promoting an anti-tumour immune response [121].

Therefore, miRNA-146a acts as a tumour suppressor, regulating various signalling pathways involved in inflammation, cell proliferation, metastasis, and immune evasion. Its downregulation or dysregulation is often observed in malignant tumours, contributing to tumour development and progression. Understanding the mechanisms by which miRNA-146a functions as a tumour suppressor may provide insight into the development of novel therapeutic strategies in cancer treatment.

### 5.2. miRNA-146a as an Oncogene

The expression of miRNA-146a has been described in many malignancy studies, where it has been found to promote tumour growth, invasion, and metastasis by affecting certain genes involved in key cellular processes. Additionally, miRNA-146a is involved in modulating inflammatory pathways and immune responses, which may contribute to cancer development and progression. Wang et al. observed increased expression of miRNA-146a in cervical cancer tissues. Transplantation of miRNA-146a into HeLa cells resulted in a significant increase in cell proliferation [122]. Further, Hu et al. found that transplantation of synthetic miRNA-146a into cervical cancer cell lines decreased IRAK1 and TRAF6 protein levels and simultaneously increased cyclin D1 expression. Although the frequency of apoptosis was not affected, more cells shifted to S-phase, suggesting an increase in proliferation rate [123]. These results were quite interesting, as downregulation of IRAK1 and TRAF6 has previously been found to have anti-tumour effects. 

A striking example of miRNA-146a as an oncogene is presented in melanoma research. Melanoma, usually formed by repeated exposure to UV radiation, is the most aggressive form of skin cancer. miRNA-146a was first analysed in melanoma in 2014, where Forloni et al. [124] found that it is elevated in this disease. Human melanoma cells proliferated faster when miRNA-146a was overexpressed, and this was attributed to miRNA-146a-mediated regulation of numb homolog protein (NUMB), a Notch pathway repressor. Further work confirmed these results and concluded that the lunatic fringe gene (LFNG) is also a direct target of miRNA-146a in melanoma cells. Melanoma recurrence was correlated with miRNA-146a levels [125]. Others have confirmed through Transwell analysis that miRNA-146a can also contribute to the migration and invasion of malignant melanoma. SMAD4 was identified as the target of miRNA-146a in these cells, and overexpression of SMAD4 could counteract the effects of miRNA on migration and invasion [126]. Overall, miRNA-146a can affect multiple signalling pathways in melanoma cells, including Notch and TGF-β. Increased levels of miRNA-146a were also observed in T-cell leukaemia and lymphoma, which are malignant T-lymphocyte tumours. Expression of miRNA-146a was increased in T-cell lines infected with human T-cell leukaemia virus type 1 (HTLV-1). This virus has been found to cause T-cell leukaemia in adults. Inhibition of miRNAs in infected cells could slow the rate of proliferation but had no effect on uninfected cells [127]. Thus, targeting this miRNA and its downstream pathways may offer potential therapeutic strategies for the treatment of oncopathology.

### 5.3. Involvement of miRNA-146a in Tumour Metastasis

Activation of NF-κB, which promotes cancer cell survival and metastatic potential, is widely recognized [128]. miRNA-146a has a tremendous effect on the course of metastasis development since it can both stimulate and inhibit this process in the body. According to the results of some studies, it can be assumed that such differences in the action of miRNA-146a in the tumour’s pathogenesis depends on the tumour’s specific localization [4,121]. 

Studies since 2007 have revealed that miRNAs play an important role in cancer cell metastasis by influencing various factors involved in cancer cell metastasis, such as epithelial–mesenchymal transition (EMT), cancer stem cell properties, etc. For the first time, in 2007, a difference in the expression level of miRNAs between breast epithelial cells and metastatic cancer cells was reported [129,130]. miRNA-146a plays a role in tumour cell development and metastasis, and its downregulation causes the activation of the NF-κB signalling pathway (Figure 3). However, in some metastatic cancers, microRNA expression levels are increased, indicating a dual role in cancer cells [131].

Studies with hepatocellular metastatic carcinoma cells have shown that a high methylation level of the miRNA-146a promoter results in reduced expression levels of miRNA-146a compared to normal cells, while increased levels result in reduced invasion levels. These studies indicate that miRNA-146a can reduce cancer cell invasion and migration [114]. Additionally, studies have shown that miRNA-146a is involved in cancer cell metastasis and invasion by affecting c-met. In colon cancer cells, the expression level of miRNA-146a is decreased, while the expression level of c-met is increased as one of the targets involved in the metastasis and invasion of colon cancer cells to the liver [132]. 

Furthermore, miRNA-146a can also partially suppress the metastatic potential of breast cancer cells, by reducing the constitutive activity of NF-κB [133]. Moreover, BRMS1 contributes to suppressing metastasis in various cancer cells by elevating miRNA-146a expression within metastatic breast cancer cells. Additionally, the tumour suppressor Merlin plays a critical role in maintaining tissue homeostasis by regulating cell contact inhibition of growth. Interestingly, miRNA-146a, along with several other miRNAs, exerts a negative regulatory effect on Merlin protein levels. As a consequence, persistent downregulation of Merlin by the expression of miRNA-146a blocks cell contact inhibition of proliferation while enhancing cell migration, tumour formation, and metastasis in A549 lung epithelial cells [134]. Metastasis and chemoresistance are interrelated processes, and chemoresistance can trigger metastasis. Several studies have shown that miRNA-146a promotes sensitivity to cisplatin or reduces resistance to cisplatin, thereby regressing metastasis in lung cancer through specific molecular targets [135]. Examples, such as miRNA-146a-5p, exemplify the complexity of the network of miRNA regulation in carcinogenesis and highlight that miRNA regulatory functions are often specific to individual cancers and their targets.

### 5.4. miRNA-146a as Regulator and the Role in Response to Cancer Therapy

One of the most important problems in the systemic treatment of cancer is multidrug resistance. Various mechanisms are involved in drug resistance. Research studies have shown that miRNAs play a significant role in drug resistance [136]. Studies using non-small cell lung cancer (NSCLC) cells have shown that increased expression levels of miRNA-146a enhance sensitivity to cisplatin (DDP)-based chemotherapy. Furthermore, a study in ovarian cancer patients demonstrated a correlation between miRNA-146a levels and drug resistance [137].

Furthermore, miRNA-146a is involved in the regulation of immune-related genes in ICI treatment. It regulates IFN-γ and perforin production in T cells, thereby improving immune-related adverse events (irAE) severity. Moreover, the miRNA-146a SNP rs2910164CC in solid tumour patients treated with checkpoint inhibitors has been highlighted as a prognostic marker to estimate the efficacy of ICI therapy [138]. In contrast, the rs2910164GC and GG genotypes were shown to be associated with lower survival in patients with RCC compared to those with the CC genotype [139]. Previously, we found a significant decrease in miRNA-146a expression level in clear cell renal cell carcinoma (ccRCC) patients with severe irAEs and the SNP rs2910164CC had a higher risk of developing severe irAEs [140]. In melanoma patients, miRNA signatures, including miRNA-146a, have previously been shown to be involved in the accumulation of bone marrow-derived suppressor cells and the development of therapeutic resistance to immune checkpoint inhibitor therapy [141].

Ongoing investigations are focusing on harnessing miRNAs as promising and modern approaches for cancer treatment, either as therapeutic agents or targets. Interestingly, research studies have demonstrated that miRNA-based treatments can enhance the sensitivity of cancer cells to specific chemotherapeutic agents. Notably, miRNA-146a has shown the ability to increase the sensitivity of diverse cancer cell types to various cancer treatments. Furthermore, a correlation has been observed between gastric cancer patients with elevated levels of miRNA-146a and improved treatment efficacy [142]. In the context of ovarian cancer, transfecting cancer cells with a miRNA-146a mimic in conjunction with paclitaxel treatment resulted in a more substantial decrease in cell viability compared to miRNA-146a or paclitaxel treatment alone [143].

Shi and colleagues found that restoration of miRNA-146a expression in NSCLC cell lines increased sensitivity to cisplatin, as evidenced by cell cycle arrest, increased levels of apoptosis, inhibited cell viability, and slower rates of cell migration [144]. Additionally, in NSCLC cells, miRNA-146a could enhance the effects of drugs targeting EGFR, including tyrosine kinase inhibitors gefitinib, erlotinib, and afatinib, as well as monoclonal antibody cetuximab [145]. 

Similar synergistic effects with cetuximab were observed in hepatocellular carcinoma (HCC) cells [146]. Additionally, in HCC cells, researchers observed a synergistic effect on proliferation and apoptosis rates when cells were transfected with miRNA-146a and treated with the natural compound ginsenoside Rh2 [147]. Recent results suggested that miRNA-146a could enhance the radiosensitivity of liver cancer cells. In terms of mechanism [148], several miRNAs have also been studied, including miRNA-146a, which may be promising candidates for biological tests predicting the radiosensitivity of patients with prostate cancer undergoing radiation therapy [149]. These initial findings strongly suggest that miRNA-146a might be useful as a prognostic biomarker and a targeted therapeutic agent in different types of cancer.

## 6. Predicted Gene Targets of miRNA-146a

miRNA-146a is a non-specific mediator that plays a role in regulating inflammation in various conditions accompanied by an inflammatory response in the body. This mediator has a large number of putative target genes, with over 300 genes described in various databases (Figure 4). Examples of target genes include cell adhesion molecule L1 (L1CAM), MMP-9, C-X-C chemokine receptor type 4 (CXCR4), vimentin (VIM), rho-activated protein kinase (ROCK1), and E-cadherin [4]. 

Prostate cancer cell lines revealed a significant decrease in ROCK1 expression upon achieving the essential levels of miRNA-146a expression. This reduction resulted in decreased migration, invasion, and proliferation processes. Furthermore, miRNA-146a directly targets MMP-9, a pivotal player in physiological and pathological processes such as cell invasion and migration [150,151]. MMP-9 is a key oncogene affecting cancer cell invasion, and high levels of MMP-9 correlate with an unfavourable cancer prognosis [121]. miRNA-146a also targets FOS, a main element of the NF-κB signalling pathway and participates in regulating the NF-κB signalling pathway, decreasing MMP-9 expression by suppressing the FOS-AP-1 pathway [152].

Three target genes of miRNA-146a, *ADORA 3*, *CYSLTR2*, and *HRH 4*, are involved in the interaction pathway of neuroactive ligands with receptors [153]. Among these, *HRH4* has the highest target score and plays a role in immunological and inflammatory processes [154,155]. Additionally, a growing body of evidence has demonstrated that HRH4 exerts important functions in cell proliferation [155]. It also affects plasmacytoid dendritic cell (PDC) migration and cytokine production [156]. 

*HNRNP D* is an oncogene and a direct target of miRNA-146a-5p. Downregulation of miRNA-146a-5p results in the overexpression of *HNRNP D*, thereby promoting malignant transformation of stromal/mesenchymal stem cells upon interaction with glioma stem cells [157]. 

*MITF* (microphthalmia-associated transcription factor), a proto-oncogenic transcription factor, is also targeted by miRNA-146a. *MITF* acts as a main controller in the progress of melanocytes’ function and survival, participation in pigmentation, and proliferation of choroid melanoma [158]. 

### Immune Genes

In response to TLR ligands and proinflammation mediators such as TNF and IL-1β, TNF production is regulated through the suppression of TRAF-6 and IRAK1 [43]. miRNA-146a is induced in human monocytes in an NF-κB-dependent manner following LPS stimulation and acts as a negative regulator of TNF production. It plays a crucial role in regulating signalling from activation of TLRs and cytokine receptors via a negative feedback loop involving downregulation of TRAF-6 and IRAK-1. Several studies have indicated that miRNA-146a has additional targets that regulate innate immunity. It is activated during a virus infection and negatively regulates the RIG-I-dependent antiviral pathway by targeting IRAK-1, TRAF-6, and IRAK-2. IRAK-2 is another miRNA-146a target that is involved in VSV-induced type I IFN production, as we have proven [51]. Furthermore, miRNA-146a is upregulated by IL-1β in alveolar epithelial cells of the lungs, leading to the negative regulation of the release of pro-inflammatory chemokines such as IL-8 and RANTES [159]. In HIV-infected microglial cells, miRNA-146a targets the proinflammatory cytokine CCL8 (MCP2), leading to a decrease in its release [160]. The identification of miRNA-146a target genes provides valuable insights into the biological processes involved in various diseases such as cancer, infectious, autoimmune, and allergic diseases, and miRNA-146a can serve as an additional molecular biomarker in the diagnosis and therapy of these diseases.

## 7. Participation of miRNA-146a in the Development of Ischemic Stroke 

Disruption of microRNA expression regulation is known to be associated with a predisposition to atherosclerotic lesions in coronary and carotid arteries, as well as the development of complications such as coronary heart disease, myocardial infarction, chronic cerebral ischemia, and ischemic stroke. Recently, an increasing number of studies have emerged analysing the microRNA regulome, which encompasses a network of regulatory elements controlling the expression of miRNAs and their respective targets. Stroke represents a significant global public health issue, contributing substantially to both mortality and morbidity worldwide. In recent years, a growing body of evidence has demonstrated that miRNAs play crucial regulatory roles in numerous physiological processes that could potentially mitigate the onset and progression of ischemic stroke. Thus, miRNAs can be significant in stroke as novel diagnostic biomarkers and potential therapeutic interventions [161,162].

The pattern of microRNA expression in brain tissue and blood has been identified to vary depending on the course of stroke and its subtypes [163]. The regulation of most neurological diseases, including ischemic stroke (IS), is closely associated with the regulation of miRNA-146a, one of the most abundant miRNAs expressed in the central nervous system [164,165,166,167,168]. For instance, the blood of patients with subacute ischemic syndrome (SI) has an elevated level of miRNA-146a [169]. Conversely, patients with acute inflammatory syndrome (ASI) exhibited the opposite result [170]. In another study, the expression of miRNA-146a was found to be significantly decreased in patients with ischemic stroke. The expression of miRNA-146a was correlated with the severity of ischemic stroke and showed a significant decrease in patients with coma compared to healthy controls [171].

Li et al. studied that miRNA-146a levels were also significantly lower in patients with ischemic stroke in the acute phase compared to patients in the subacute phase and healthy control groups [169]. These findings have been corroborated by genetic studies demonstrating a significant association between miRNA-146a polymorphisms and the risk of ischemic stroke [172,173]. Zhou et al.’s work illustrated that miRNA-146a suppression exerts a protective effect on nerve cells and targets pro-apoptotic genes to reduce apoptosis [174].

Stroke-induced oligodendrogenesis was also found to be enhanced by miRNA-146a [175]. We can assume that miRNA-146a has the potential to be a promising therapeutic target during the acute and recovery phases of ischemic stroke, as highlighted in these studies [176].

In young patients, pilocytic astrocytoma (PA) is the most common tumour of the central nervous system [177]. miRNA-146a is significantly activated in the brains of children with PA and negatively regulates aging-related inflammatory responses and the cell cycle through the NF-κB and extracellular regulated protein kinase/mitogen-activated protein kinase (ERK/MAPK) pathways [178].

The relationship between miRNA-146 polymorphic variants and the development of ischemic stroke is also under investigation. Most of the research has been conducted on Chinese populations. A study by Zhu et al. [172] demonstrated that the miRNA-146a C allele and CC genotype could increase the risk of stroke in the Han population in northern China. Qu et al. [179] reported that miRNA-146a (rs2910164) could be used as a diagnostic marker for the prognosis of stroke in the Asian population. Huang et al. [180] claimed that miRNA-146a (rs2910164) was significantly correlated with stroke in the Han population in Shenzhen, China.

In 2018, a group of authors from Guangxi University (China) discovered that the expression of miR-146b in serum significantly increased within 4 h of stroke onset compared to the control group. However, they did not observe differences in the expression of miR-146b based on the subtype of stroke [181].

In another study, Young Joo Jeon et al. investigated susceptibility to stroke and silent brain infarction (SBI). They examined miRNA-146 polymorphisms in 678 patients using restriction fragment length polymorphism PCR (RFLP) analysis. The results revealed that the G allele of the miRNA-146a polymorphism was associated with an increased risk of IS in the Korean population. This study marks the first report of the association between stroke, SBI, and the miRNA-146a polymorphism C > G in the Korean population [173].

In addition, a meta-analysis conducted in 2019 investigated the association between miRNA-146a rs2910164 genetic polymorphisms and the risk of IS in the Chinese population. A total of ten eligible studies, comprising 4251 patients with IS and 5812 controls, were ultimately included. The results showed that the pooled odds ratio (OR) for IS of the rs2910164 G allele was 1.23 (95% CI: 1.03–1.46, *p* = 0.022) compared to the wild-type C allele under a recessive model, with high heterogeneity (I² = 56.2%, *p* = 0.015) [182]. 

Considering the complex and extended process of atherosclerotic plaque formation, characterized by high cellular heterogeneity, it becomes possible to assume that joint changes in the expression of various miRNAs occur during the development of atherosclerosis, or alterations in the cellular composition of arteries lead to the formation of microRNA co-expression modules. Determining microRNA profiles allows not only the confirmation of events but also the identification of the stroke’s timing, differentiation between stroke subtypes, and even prognostication. These highlights provide insights to identify important new targets for enhancing screening methods as well as therapeutic strategies for ischemic brain injury and neurodegenerative diseases [183,184,185].

## 8. Conclusions

To date, numerous studies have demonstrated the role of miRNAs in the pathogenesis, diagnosis, and treatment of various diseases. However, further research in this direction is warranted, as miRNAs hold significant promise in clinical practice.

miRNA-146a affects multiple target genes through signalling pathways, thereby regulating gene expression and, consequently, protein expression. The exact mechanism of this effect is not yet fully understood, but it has been demonstrated, for example, that miRNA-146a influences TRAF6, leading to reduced expression of NF-κB and participation in inflammatory reactions within the body. Furthermore, miRNA-146a is involved in the regulation of TLRs that recognize PAMPs.

Furthermore, there is significant evidence regarding the substantial contribution of miRNA-146a to the pathogenesis of malignant neoplasms. It regulates processes such as cell proliferation, the development of metastases, and even the response to ongoing therapies, including radiation and chemotherapy. There is reliable knowledge that the concentration of miRNA-146a in the biological fluids of patients differs significantly from its concentration in the biological fluids of healthy individuals. Consequently, it can serve as a valuable biomarker across various fields of medicine.

Thus, miRNA-146a holds significant potential in personalized medicine, emphasizing the crucial significance of studying the exact regulatory mechanisms in which miRNA-146a is involved.

## Figures and Tables

**Figure 1 ijms-24-12767-f001:**
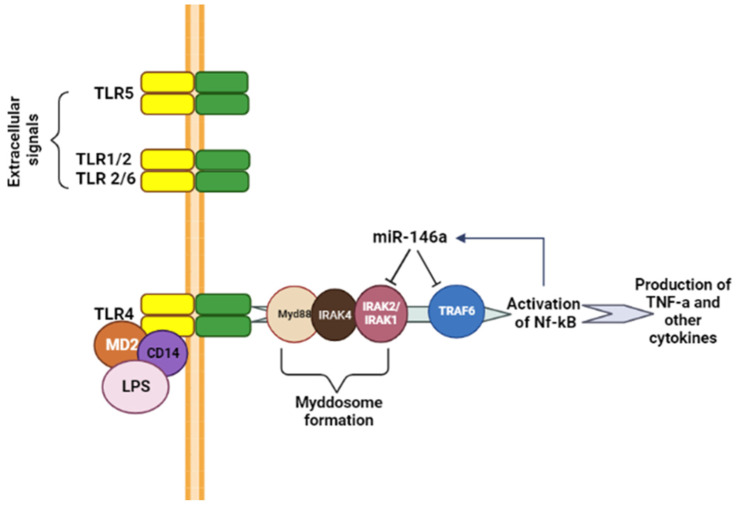
miRNA-146a targets involved in the MyD88-dependent signalling pathway.

**Figure 2 ijms-24-12767-f002:**
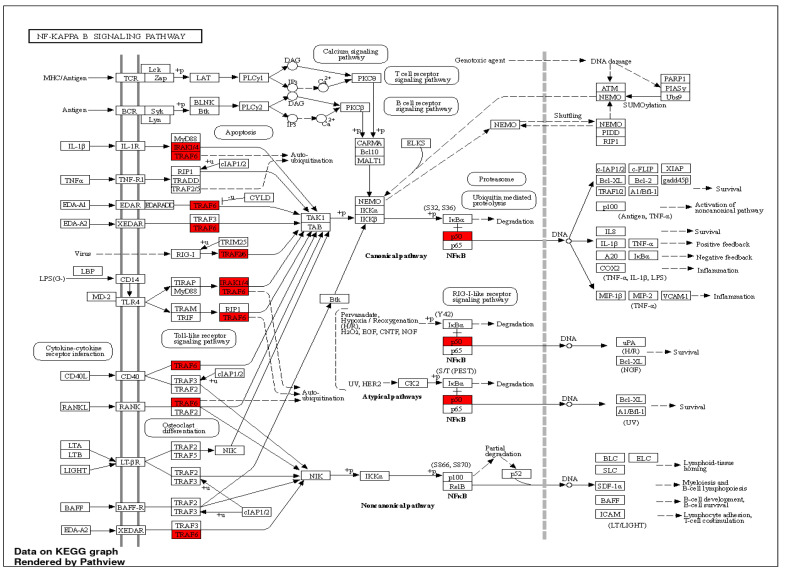
Target genes of miRNA-146a validated for the TLR signalling pathway (indicated in red), based on the KEGG database.

**Figure 3 ijms-24-12767-f003:**
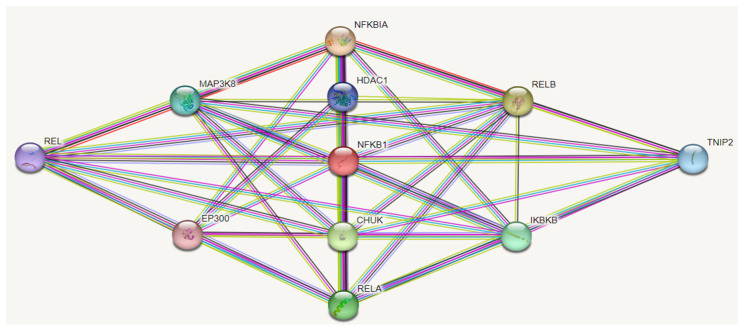
miRNA-146a targets involved in NF-kB signalling pathway.

**Figure 4 ijms-24-12767-f004:**
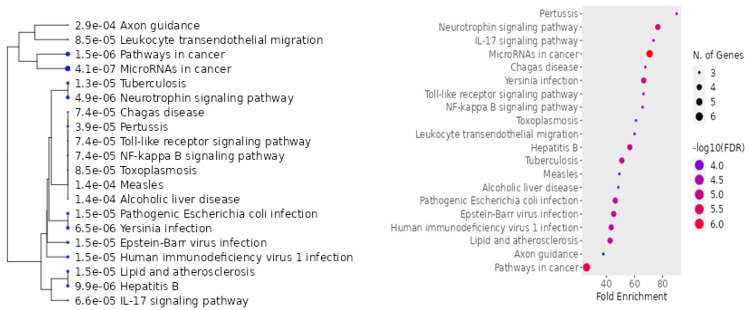
A dot-plot was performed using ShinyGO resource to represent the outcomes of KEGG pathway enrichment analyses carried out on miRNA-146a and its validated targets.

## Data Availability

Data sharing is not applicable to this article as no datasets were generated or analysed during the current study.

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
