# Peer review of "MiRNA-146a—A Key Player in Immunity and Diseases"

_ijms, 2023, doi:10.3390/ijms241612767_

Round 1

Reviewer 1 Report

The review reports recent discoveries unveiling the significant role of miRNA-146a in regulating immune system and oncogenesis by regulating several signaling pathways. The article is well organized and figures quite attractive and fitting with the content. However, some issues should be improved:

Please update, shorten or rewrite paragraph "2. MiRNA-146a - function, role, involvement in innate immunity, development of inflammatory responses, and hematopoiesis" since it contains old references, not corresponding to the aim of this review stated in the abstract "In this review, we focus on recent discoveries". The only recent reference of the paragraph is dated 2019 (Ref 21). Usually a review should refer to the last 5-years update. 

Please check acronyms along the whole text, including different "k" of NF-kB. Some extensive names followed by acronyms are repeated several times (non only at first appearance), leading to a very confusing appearance.

Minor:

Line 55: sentence should begin with "miRNA-146a" instead of "it", since in this paragraph miRNA has not been named yet.

Line 188: "immune immune response".

Line 192: please, use acronym RA 

Line 265: IRAK1 please, explain acronym previously (line 236)

Line 270: NF-kB please, explain acronym previously

Line 373, 578, 611, 646 please check

Language is quite adequate and generally the manuscript is well written. However, some editing mistakes and recurrent full name-acronym combos lead to confused appearance. The manuscript should be checked throughout.

Author Response

Dear Reviewer1,

Thank you for a careful analysis of the article "MiR-146a - a key player in immunity and diseases" and valuable comments. The article has been revised according to the recommendations and comments:

  1. We have updated and shortened the paragraph devoted to miRNA-146a - function, role, involvement in innate immunity, development of inflammatory responses, and hematopoiesis
  2. We have checked acronyms along the whole text and corrected them all, brought the names to uniformity, and italicized the name of genes.
  3. In line 55 (in the revised version of the article it is the line 52) the sentence begins with «miRNA-146a» instead of «it».
  4. In line 188 (in the revised version of the article it is the line 149) we have deleted an extra word «immune».
  5. From line 151 we now use acronym RA throughout the text.
  6. We have explained acronym IRAK1 at the first mention in the text on line 79.
  7. We have explained acronym NF-κB at the first mention in the text on 58.
  8. We have checked and corrected lines 373,578,611,646.

Besides, we have done mistakes correction and deleted recurrent full name-acronym combos. The manuscript has been checked throughout.

Reviewer 2 Report

This review is nicely written about the miRNA and its translational opportunities. It's a vital player in Stroke associated inflammation as well as cancer-associated infarcts too. I would recommend that authors can write the role of MiR-146a in the pathology of stroke and cancer. Patients with stroke can develop cancer, and those with brain cancer can quickly develop stroke. Hence it's worth writing an update on miRNA for stroke and cancer.

Authors can take a reference of below articles:

Coding and non-coding nucleotides': The future of stroke gene therapeutics

The expression of microRNA 146a in patients with ischemic stroke: an observational study

MicroRNA-146a Is a Wide-Reaching Neuroinflammatory Regulator and Potential Treatment Target in Neurological Diseases

Regulatory effects of miR-146a/b on the function of endothelial progenitor cells in acute ischemic stroke in mice

Association of the miR-146amiR-149miR-196a2, and miR-499 Polymorphisms With Ischemic Stroke and Silent Brain Infarction Risk

Genetic variants of miR-146a and miR-499 and risk of ischemic stroke in the Chinese population: a meta-analysis and trial sequential analysis

Association of miR-146a, miR-149, miR-196a2, miR-499 gene polymorphisms with ischemic stroke in a Chinese people

We appreciate the authors that have been writing nicely by authors about this topic. The authors have written nicely about this topic which contains nice translational views. The authors should go through minor English and Grammer Revisions after topic addition as per the minor revisions suggested.

Author Response

Dear Reviewer2,

Thank you for a careful analysis of the article "MiR-146a - a key player in immunity and diseases" and valuable comments. The article has been revised according to all your recommendations and comments:

  1. According to your comment, we have analyzed the role of miR-146a in the pathology of stroke and cancer, added a new paragraph devoted to participation of miRNA-146a in the development of ischemic stroke (from line 648 onwards).
  2. We have performed editing of English language throughout the text.
